# The Antifungal Activity of Ag/CHI NPs against *Rhizoctonia solani* Linked with Tomato Plant Health

**DOI:** 10.3390/plants10112283

**Published:** 2021-10-25

**Authors:** Ameena A. Al-Surhanee, Muhammad Afzal, Nahla Alsayed Bouqellah, Salama A. Ouf, Sajid Muhammad, Mehmood Jan, Sidra Kaleem, Mohamed Hashem, Saad Alamri, Arafat Abdel Hamed Abdel Latef, Omar M. Ali, Mona H. Soliman

**Affiliations:** 1Biology Department, College of Science, Jouf University, Sakaka 2014, Saudi Arabia; amaserhani@ju.edu.sa; 2Institute of Soil and Water Resources and Environmental Science, College of Environment and Resource Sciences, Zhejiang University, Hangzhou 310058, China; 3Islamic Girls School and College, Parachinar 26301, KPK, Pakistan; 4Biology Department, Science College, Taibah University, Medina 42317-8599, Saudi Arabia; Nahla.B@hotmail.co.uk; 5Botany and Microbiology Department, Faculty of Science, Cairo University, Giza 12613, Egypt; SaOuFeg@yahoo.com; 6Institute of Crop Science, College of Agriculture and Biotechnology, Zhejiang University, Hangzhou 310058, China; msajid1772@gmail.com (S.M.); mehmoodjan89@gmail.com (M.J.); 7Ocean College of Science and Engineering, Zhejiang University, Zhoushan 316021, China; kaleemsidra85@yahoo.com; 8Department of Biology, College of Science, King Khalid University, Abha 61413, Saudi Arabia; drmhashem69@yahoo.com (M.H.); amri555@yahoo.com (S.A.); 9Department of Botany and Microbiology, Faculty of Science, Assiut University, Assiut 71516, Egypt; 10Botany and Microbiology Department, Faculty of Science, South Valley University, Qena 83523, Egypt; moawad76@gmail.com; 11Department of Chemistry, Turabah University College, Turabah Branch, Taif University, P.O. Box 11099, Taif 21944, Saudi Arabia; om.ali@tu.edu.sa; 12Biology Department, Faculty of Science, Taibah University, Al-Sharm, Yanbu El-Bahr, Yanbu 46429, Saudi Arabia; monahsh1@gmail.com

**Keywords:** *Rhizoctonia solani*, nanoparticles, plant defense, chitosan, antioxidants

## Abstract

Pathogenic infestations are significant threats to vegetable yield, and have become an urgent problem to be solved. *Rhizoctonia solani* is one of the worst fungi affecting tomato crops, reducing yield in some regions. It is a known fact that plants have their own defense against such infestations; however, it is unclear whether any exogenous material can help plants against infestation. Therefore, we performed greenhouse experiments to evaluate the impacts of *R. solani* on 15- and 30-day old tomato plants after fungal infestation, and estimated the antifungal activity of nanoparticles (NPs) against the pathogen. We observed severe pathogenic impacts on the above-ground tissues of tomato plants which would affect plant physiology and crop production. Pathogenic infection reduced total chlorophyll and anthocyanin contents, which subsequently disturbed plant physiology. Further, total phenolic contents (TPC), total flavonoid contents (TFC), and malondialdehyde (MDA) contents were significantly increased in pathogen treatments. Constitutively, enhanced activities were estimated for catalase (CAT), superoxide dismutase (SOD), and ascorbate peroxidase (APX) in response to reactive oxygen species (ROS)in pathogen-treated plants. Moreover, pathogenesis-related genes, namely, chitinase, plant glutathione S-transferase (*GST*), phenylalanine ammonia-lyase (*PAL1*), *pathogenesis-related protein* (*PR12*), and *pathogenesis-related protein* (*PR1*) were evaluated, with significant differences between treated and control plants. In vitro and greenhouse antifungal activity of silver nanoparticles (Ag NPs), chitosan nanoparticles, and Ag NPs/CHI NPs composites and plant health was studied using transmission electron microscopy (TEM) and Fourier transform infrared (FTIR) spectrophotometry. We found astonishing results, namely, that Ag and CHI have antifungal activities against *R. solani*. Overall, plant health was much improved following treatment with Ag NPs/CHI NPs composites. In order to manage *R. solani* pathogenicity and improve tomato health, Ag/CHI NPs composites could be used infield as well as on commercial levels based on recommendations. However, there is an urgent need to first evaluate whether these NP composites have any secondary impacts on human health or the environment.

## 1. Introduction

*Rhizoctonia solani* (Kuhn) is a ground-dwelling plant pathogenic fungus with a broad-spectrum of hosts and global spread, causing plant diseases like collar rot, root rot, damping off, and wire stem [1]. The pathogen is identified by genetically segregated inhabitants called “anastomosis groups” [2]. *R. solani* is the smallest fungus known to originate from “damping-off”, and can live in soil without a host for several years. The main reason behind this survival is the formation of sclerotia, a dense pile of toughened fungal mycelia 1–3 mm in diameter including food stocks [3]. Its broad host range and ability to compose sclerotia make this pathogen very challenging to manage. This pathogen causes disease in several important vegetables and crops including corn, tomato, potatoes, cereals, sugar beet and cucumber.

Tomato (*Solanum lycopersicum* L.) is the second-most commonly consumed vegetable crop worldwide, after potato [4]. Several pathogens like fungi, bacteria, nematodes, and viruses can infect tomato plants. [5]. Among fungal pathogens, *R. solani* is the most damaging for tomato plants [6]. Although there are techniques to stop the spread of these pathogens, chemical fungicides are generally used. The role of these fungicides has been questioned due to their lethal effects on nontarget organisms [7]. In contrast, it has been reported that beneficial bacteria can inhibit phytopathogenic fungi by inducing cellular defense responses in plants [8].

In adverse environments, plants have to evolve various defense mechanisms that enable them to avoid tissue damage when pathogens attack. Systemic acquired resistance (SAR) and induced systemic resistance (ISR) are involved in plant systemic immunity. SAR is a salicylic acid (SA)-mediated, broad-spectrum, disease-resistance response of plants to pathogens, usually triggered by necrotrophic fungi and bacteria. In contrast, ISR is the response of beneficial microorganisms such as plant growth-promoting rhizobacteria (PGPR), which canregulate jasmonate (JA)- and ethylene (ET)-dependent signaling pathways, in turn enhancing plant immunity rather than directly activating its defenses [9]. There is obvious evidence for the systemic activity of defense-related enzymes such as superoxide dismutase (SOD), catalase (CAT), and ascorbate peroxidase (APX), as well as the expression of defense-related genes, e.g., *pathogenesis-related protein* (*PR-1*), a salicylic acid (SA) marker gene, *PR-3*, chitinase encoding gene, *glutathione-S-transferase* (*GST*), and *defensin encoding gene* (*PR12*) enhanced by *Bacillus* sp. in soybean, tomato, and *Arabidopsis thaliana* [10,11,12,13]. *Phenyl ammonia-lyase* (*PAL*) is a key enzyme involved in phenylpropanoid metabolism, leading to the production of defensive compounds (lignins, coumarins, flavonoids, and phytoalexins) [9].

Nanoparticles (NPs) have unique physico-chemical, biological, and optical properties, and are utilized as antimicrobials in various disciplines. The implementation of nanotechnology has revealed huge possibilities in managing fungi and pathogenic bacteria, especially in the agriculture and food sectors. Despite the antimicrobial and antipathogenic activities of these NPs, their mechanisms are not well understood. However, the utilization of silver nanoparticles (Ag NPs) as an antifungal agent has been broadly validated through scientific study. Indeed, Ag NPs can be helpful in plant disease control against pathogenic fungi [14]. In a recent study, the effect of Ag NPs on *R. solani* groups that contaminate cotton plants was assessed [15]. Ag NPs generate reactive oxygen species (ROS), especially superoxide radicals (O^−2^) and hydroxyl radicals (OH), that destroy the cell [16]. The biological activity of chitosan nanoparticles (CHI NPs) in foodborne bacteria has been correlated with particle size, mass, and PH. Many studies have supported the efficacy of particles made from materials such as silver, copper, and metal ions with CHI NPs in the management of pathogenic bacteria [17].

Methods for detecting and quantifying *R. solani* in soil are highly laborious and time-consuming, involving the use of soil baiting methods that are often inefficient in detecting the pathogen [18]. Furthermore, low population densities of *R. solani* in soil and a lack of selective isolation media for the species make quantifications difficult and unreliable. In the last decade, several conventional or real-time quantitative polymerase chain reaction (qPCR) assays have become established tools for rapidly quantifying fungal pathogens for *R. solani* at low detection limits in both soil and infected plant tissues [19]. In the present study, we assessed the antifungal activity of Ag/CHI NPs against *R. solani* in a greenhouse setting. We further examined plant health and defense by molecular methods when soil was infested with *R. solani*.

## 2. Results

### 2.1. R. solani Is the Cause of Root and Crown Rot Diseases in Tomato Plants

*R. solani* is a plant pathogenic fungus with a wide host range and worldwide distribution. It is a phytopathogen that attacks tomatoes cultivated under greenhouse conditions, causing root and crown rot diseases. In this study, the fungal activity of *R. solani* was tested against tomato (*Solanum lycopersicum* L.).

According to our observations, fungus applied treatment (P) was mainly affected based on the levels of disease severity (DS) of up to 92%, while all other treatments showed less than 10%DS values, illustrating severe disease in the P treatment (Figure 1A). Moreover, plant physiology was also severely affected in pathogen-treated plants. Accordingly, significant differences were observed for plant height (PH) among treatments, where minimum PH was monitored for P and P + NC, revealing the effects of the fungus on the treated plants compared to control plants (Figure 1B). Shoot fresh weight (SFW) and shoot dry weight (SDW) also followed the same pattern as PH, where by minimum shoot weights were observed for P and P + NC treatments for both traits, showing the impact of the fungus on plant physiology (Figure 1C,D).

Very few differences were observed among roots traits (root fresh weight and root dry weight) among all treatments, indicating low impact of the fungus on root physiology (Figure 1E,F). In contrast, other traits like the number of leaves per plant and leaf area revealed significant impacts of the fungus for P and P + NC treatments (Figure 1G,H). Overall, the experiments demonstrated a direct effect on PH, SFW, SDW, number of leaves, and leaf area under P and P + NC treatments compared to C and NC treatments.

### 2.2. Physiological Characterizations of Tomato Plants

Physiological characterization of the experiments showed a significant decrease in growth parameters for shoot areas of plant, i.e., plant height (PH), SFW, SDW, number of leaves, and leaf area under P and P + NC treatments. In contrast, no apparent effects on the root physiology of tomato plants were observed.

There was almost no difference among plants after 14 and 29 days of treatment, indicating that the fungus has long-lasting effects on plant physiology, and that the impact persists after emergence (Figure 2).

As expected, the lowest PH was observed in the case of P and P + NC treatments, which showed the effects of the fungus on the stem and shoots of the plants; however, plant physiology appeared to be normal for C and NC treatments. Overall, the experiment demonstrated a direct effect of the fungus on the aerial parts of the plants under P and P + NC treatments compared to C and NC treatments (Figure 2). Apart from that, we also observed nonsignificant changes in roots traits in almost all treatments applied in the experiment.

### 2.3. R. solani Significantly Impacted Plant Photosynthetic Pigments

We also checked whether a fungal intrusion had any effects on the photosynthetic pigments. To this end, we evaluated total plant pigments in terms of chlorophyll and anthocyanin contents. Highly significant differences were observed in terms of total chlorophyll among all treatments. As expected, fungal-treated plants had low chlorophyll contents compared with control plants. In the case of controls, high contents were observed for NC and C treatments, i.e., 55.12 and 50.32 (µg g^−1^ FW). However, fungal application resulted in reduced chlorophyll contents, i.e., 44.81 and 39.27 (µg g^−1^ FW), in the case of P + NC and P treatments, demonstrating a significant decrease in the content upon exposure to a fungal environment (Figure 3A)

Highly significant differences were observed in terms of total anthocyanin among all treatments. As expected, according to chlorophyll content, fungal treated plants had low anthocyanin contents compared with control plants. In the case of controls, high contents were observed for NC and C treatments, i.e., 1.34 and 1.22 (µg g^−1^ FW). However, fungal application led to reduced anthocyanin contents, i.e., 1.10 and 0.70 (µg g^−1^ FW), in the case of P + NC and P treatments, demonstrating a significant decrease in the content upon exposure to a fungal environment (Figure 3B,C).

### 2.4. Influence of the Fungus on Different Biochemicals

In the second part of this study, biochemical analyses of the fungal treated plants were compared with those of nontreated plants and their effect on malondialdehyde (MDA), total phenolic contents (TPC), total flavonoid contents (TFC), and total protein contents. Significantly high MDA was detected in the case of P + NC treatment, i.e., 92.41 (nmol g^−1^ FW). In the case of P, 86.16 (μmol g^−1^ FW) MDA was recorded, illustrating a strong influence of the fungus on this chemical. On the other hand, control treatments recorded 34.01 and 33.56 (nmol g^−1^ FW) of MDA in cases of NC and C, respectively, illustrating the influence of the fungus between treatments (Figure 4A).

Total phenolic content (TPC) is another biochemical factor in plants with redox, i.e., antioxidant properties. Following the same pattern as MDA, the highest TPC, i.e., 42.63 (μmol g^−1^ FW), was detected in P + NC plants. A TPC of 38.92 (μmol g^−1^ FW) was recorded in the case of P treatment, indicating the positive response of the chemical to the presence of the fungus. However, a significantly lower TPC, i.e., 19.80 and 18.60 (μmol g^−1^ FW), was observed regarding NC and C treatments, respectively (Figure 4B). These results indicated that phenolic compounds were produced in high quantities when tomato plants had been exposed to the fungus.

In addition to TPC, TFC is an essential biochemical that can be produced in plants upon exposure to stress. Interestingly, all three chemicals tested in this experiment showed consistently high values in the case of fungal applications. In this regard, the highest TFC value, i.e., 121.07 (μmol g^−1^ FW), was detected in P + NC plants. A TFC of 117.30 (μmol g^−1^ FW) was recorded in the case of P treatment, indicating a positive response of the chemical following application of the fungus. However, a significantly lower TFC, i.e., 60.25 and 55.07 (μmol g^−1^ FW), was estimated regarding NC and C treatments, respectively (Figure 4C).

In the next step, we examined the total protein content of the treatments, and interestingly, achieved the anticipated results, i.e., highly significant differences were observed among all treatments in terms of protein production. The highest protein values were detected for nonfungal treatments NC and C, i.e., 31.10 and 28.20 (nmol g^−1^ FW), respectively. Accordingly, low proteins values were recorded in P and P + NC, i.e., 16.50 and 19.75, respectively (Figure 4D).

### 2.5. Influencing Antioxidants

Regarding antioxidants, we detected the highest value for fungal treatments like P + NC and P, where SOD activity was 67.13 and 59.29 (µg g^−1^ min^−1^ protein). The least SOD activity, i.e.,26.42 and 22.35 (µg g^−1^ min^−1^ FW), was recorded with NC and C, respectively (Figure 4E).

The highest CAT activity was detected in P and P + NC treatments, i.e., 95.0 and 88.27 (µg mg^−1^ min^−1^ protein), respectively. Relatively low CAT activity was observed for CK (41.64) and NC (38.18) (µg mg^−1^ min^−1^ protein) treatments, respectively illustrating a potential increase in the enzyme in the case of P and P + NC on account of damage caused by the fungus (Figure 4F). Regarding APX activity, the highest value, i.e., 27.41 (µg g^−1^ min^−1^ protein) was detected in the case of fungal treatment P + NC treatment, followed by P, i.e., 23.30 (µg g^−1^ min^−1^ protein), showing an increase in the APX activity after the application of the fungus.

Significant differences were observed for APX activity while comparing fungus application and control. In the case of NC and C APX, these values were 14.51 and 13.68 (µg g^−1^ min^−1^ protein), respectively (Figure 4G).

### 2.6. Expression Levels of Defense-Related Genes

Additionally, pathogenesis-related genes are of the utmost importance in plants, serving to significantly bolster their defense mechanisms against a wide range of pathogens, including fungi. Therefore, in the third part of the current study, we evaluated the relative gene expression of defense-related genes to examine changes in the defense mechanisms of the plants. In this regard, the expression of five genes, namely, *Chitinase*, *GST*, *PAL1*, *PR12*, and *PR1*, was evaluated.

As expected, all genes showed significant expression levels following P and P + NC treatment, illustrating the activation of defense-related machinery after the application of these treatments. In summary, chitinase (*PR3*) GE was significantly high (18.29 and 15.9) for P and P + NC treatments, while relatively low expression was observed for the gene following C and NC treatments (Figure 5A). *GST* GE levels were also significant (9.13 and 6.29) for P and P + NC treatments, while other treatments showed relatively low chitinase GE levels (Figure 5B).

*PAL1* GE levels were also significantly high (9.13 and 6.29) for P and P + NC treatments, while other treatments showed relatively low levels for the gene (Figure 5C). Accordingly, defensin (*PR12*) GE levels were significantly high (22.16 and 23.81) for P and P + NC treatments, while other treatments showed relatively low and non-significant GE levels (Figure 5D). Finally, *AFPRT* (*PR1*) levels were also significant (9.13 and 6.29) for P and P + NC treatments, while other treatments showed relatively low *PR1* GE levels (Figure 5E).

### 2.7. Correlation of Fungal Disease with Physiological Characters

It is often useful to determine the relationship between two quantitative variables and measure their performance. In such a case, correlation is a prime tool for obtaining an accurate idea of the working capacity and strength of that relationship with available statistical data. Therefore, we conducted a correlation analysis which further confirmed our results (Figure 6).

Concurrently, we also performed a PCA analysis to identify the relationships among the variables under different treatments. Correlations between variables were detected via biplot analysis, where an acute angle indicates a positive relationship, and an obtuse angle a negative one; a right angle indicates no correlation. Altogether, both PCs explain 96.3% of the total variance of all the analyzed variables, where PC1 has the largest variance due to its orthogonal transformation. According to the PCA calculated for the experiments, PC1 explains 79.4.6% of the total variance of the variables, while the second factor (PC2) explains about 16.9% (Figure 7).

### 2.8. Evaluation of the Antifungal Properties of Ag and CHI through FTIR Spectroscopy

We applied other methods to further confirm our results regarding the antifungal properties of Ag and CHI NPs. Studies have shown that silver nanoparticles have the potential to serve a santifungal agents by destroying membrane integrity. Chitosan has been shown to have antifungal activity; its positive charge enables it to interact with negatively charged phospholipid components in the fungi membrane. Therefore, the antifungal activity of Ag/CHI NP was estimated using an agar plate technique with different concentrations of NPs in vitro, in order to determine whether it could inhibit the radial mycelial growth of *R. solani*. We found astonishing results, namely, that Ag and CHI have antifungal activityagainst *R. solani.* Using transmission electron microscopy (TEM) on the assay containing Ag and CHI NPs, we were able to check the adhesion of these NPs in the cells (Appendix A).

In FTIR, absorption bands of Ag NPs were observed at 3285.39 and 1394.11 cm^−1^. The vibrational bands corresponded to bonds, such as those of alcohols (−O−H), amine (=N−H) stretching, alkenes groups (>C = C), flavonoids, and amines (−NH_2_). In the AgNP spectrum, anew band at 1632.61 cm^−1^ suggested possible anamide (N−H) bending, while a band at 1394.11 cm^−1^ indicated the presence of an amine group. Furthermore, a band at 1026.97 cm^−1^ confirmed the presence of ether and ester functional groups (Appendix A).

Fourier transform infrared (FTIR) spectroscopy was performed to determine functions such as alcohol, amine, phenol, and carbonyl groups. FTIR absorption bands for chitosan nanoparticles at 1382.49 cm^−1^, 1528.33 cm^−1^ revealed C–H bending and N–O stretching, respectively. Absorption bands at 1016.33 cm^−1^ and 3158.51 cm^−1^ confirmed the presence of −C=O and N–H groups, respectively (Appendix A).

FTIR absorption bands at 3001.56 and 1453.51 cm^−1^ were attributed to bonds, e.g., of alcohols (−O−H), amines (=N−H), alkenes groups (>C = C), flavonoids, and amines (−NH_2_), all of which are in the range of 718–3100 cm^−1^. In the AgNP spectrum, anew band at 1738.70 cm^−1^ suggested a possible new C = O group (a ketone or an aldehyde). Finally, we also observed C−H and C−N stretching at 2917.50 cm^−1^, N−H bending at 1794.57 cm^−1^, N–H angular deformation in CO NH plane at 1453.51 cm^−1^, and C–O–C band stretching at 1158.78 cm^−1^ (Appendix A).

## 3. Discussion

*Rhizoctonia solani* is an important soil-borne fungus that infects tomato (*Solanum lycopersicum* L.), with symptoms typically manifesting at the seedling stage. Lesions on infected plant stems are mostly irregular in shape, water-soaked, brown, and with a sunken appearance [20]. *R. solani* causes significant pre- and post-emergence damping-off, characterized by the inhibition of seed germination, shoot elongation, and ultimately, the digestion of the root and hypocotyl of the plant species [21]. The disease caused by *R. Solani* has become an urgent problem to be solved. The use of chemical fungicides against *R. solani* has been limited because of drug resistance, environmental pollution, and restrictions on their use in organic agriculture [22]. The present research reports evaluations of the above-ground symptoms and the impact of *R. solani* on the developing shoot systems of tomato plants. We observed highly significant values for DS following P treatment (Figure 1). Moreover, significantly short and wilted plants were observed for shoot parameters in our experiments following P and P + NC treatments, with no significant change in root physiology, demonstrating the impact of the disease and the pathogen in predominantly the above-ground parts of plants (Figure 2).

### 3.1. Estimation of Antioxidants

In plant-pathogen communication, ROS plays a double role. It can activate a host–disease resistance response, even though the accumulation of active oxygen beyond a certain amount will also damage the host cells. Fortunately, many antioxidant enzymes are used to scavenge ROS in plants, such as SOD, APX, and CAT [10]. PAL, on the other hand, can synthesize phenols, lignin, and other substances that are associated with disease resistance by catalyzing key enzyme phenylalanine, which plays an important role when plants are attacked by pathogens. Phenolic compound oxidation and anthraquinone synthesis properties have been reported for polyphenol oxidase (PPO), which can deactivate pathogens [23]. Therefore, the ROS compounds in our observations were significantly abundant in plants following P and P + NC treatments, demonstrating that the application of pathogens caused severe damage. Yet, disease resistance catalyzed a counter-attack, assisting the plants (Figure 4).

### 3.2. Differential Responses of Microorganisms toward Different Tissues

Plants often manifest variegated responses in different tissues. Therefore, SFW and SDW were tested for tomato infected with *R. solani*. The pathogen was shown to significantly reduced plant growth, as measured by fresh root and shoot weight, and to affect root architecture, albeit not as much as with above-ground tissues (Figure 2). Plants infected with *R. solani* showed thicker roots in comparison to uninfected controls. The most significant root diameter of infested plants was inverse correlated to an increased size of the whole plant. This result is in disagreement with the general statement that plants regulate the size of their organs concerning total size/length [24].

Similarly, decreased shoot parameters could be ascribed to a hormonal imbalance, that would also account for reduced photosynthetic activity. Since the total weight and height of infected plants were significantly reduced, the total height was expected to be lower than that of healthy plants. This difference would account for the reduced growth of diseased plants, as measured by the plant weight. In line with this hypothesis, we observed a significant decrease in the abundance photosynthetic pigments such as chlorophyll and anthocyanin (Figure 3).

Plant components can be polar or nonpolar. Phenolic components are important plant constituents with redox properties which are responsible for their antioxidant activity. In contrast, flavonoids are secondary metabolites with antioxidant activity, the potency of which depends on the number and position of free OH groups (Figure 4) [25]. We observed a significant increase in the concentration of these compounds in response to the presence of pathogens, illustrating a passive response to pathogen attack.

### 3.3. Induction of Defense Related Genes

The induction of plant defenses is a new biological method for controlling plant diseases [22]. SAR and ISR are involved in plant systemic immunity. SAR is a salicylic acid (SA)-mediated, broad-spectrum, disease-resistance response of plants to pathogens which is usually triggered by pathogenic bacteria [22].

SA is an important pathogenic signal molecule which can also induce the expression of *PR* genes and enhance plant resistance to pathogens [26]. Beneficial microbes such as ISR can activate the SA-dependent signaling pathway. When the SA-dependent signaling pathway is activated, levels of PR proteins, such as chitinase and defensin, will increase. It has been reported that *B. subtilis* can increase the expression of PR proteins such as chitinase and induce tomato systemic resistance against soft rot disease [10]. In the present study, the fungus was shown to activate the SA-dependent signaling pathway by significantly elevating the expression of chitinase, *GST*, *PAL1*, defensin and *PR3* in P and P + NC treatments (Figure 5).

*PR-1*, the salicylic acid (SA) marker gene, is a crucial regulator of SAR which may indicate early defense response in plants. Indeed, increasing plant resistance is often associated with *PR-1* induction and SA content accumulation [27]. In the present study, tomato plants infected with *R. solani* showed upregulation of *PR-1*, with a relative expression level six times higher than that of control plants. Interestingly, tomato plants which had undergone P treatment showed the highest expression level, followed by those which had undergone P + NC treatment. Consequently, we propose that infected plants may produce an elicitor metabolite molecule that induces the immune defense system, resulting in SAR activation. The upregulation of *PR1* could be related to SAR activation and ISR status [28].

*PR-3*, an encoding a chitinase enzyme that catalyzes the hydrolysis of chitin, is a fungicide that protects plants from fungal infestations by inhibiting pathogengrowth. The ability of some microbial strains to inhibit *R. solani* growth may be due to differences in mycoparasitism activity through the secretion of enzymes, e.g., chitinase, that degrade the fungal cell wall [29]. In terms of mycoparasitic action, chitinase is one of the most critical extracellular lytic enzymes. In the present study, tomato plants treated with *R. solani* exhibited significant upregulation of the *PR-3* gene. The obtained results revealed the role of *PR-3* in increasing plant resistance against fungal infection [30].

A previous study showed that *R. solani AG3PT* induces systemic defense responses in sprouts [31]. Based on the present findings, which revealed pathogen colonization in the roots in early phases of plant–pathogen interactions, an activation of defense responses in roots was to be expected. In this regard, defense responses in tomatoes were assessed by monitoring the expression of common defense-related genes associated with the SA (*PR-1*, *PR-3*, *GST*, and *PAL*) signaling pathway. Regarding the analyzed genes, *R. solani* resulted in the overexpression of genes *PR-1* and *PR-3*. Both genes were found to be upregulated in tomato roots in response to the pathogen *Fusarium oxysporum* and *PR-1* in response to *R. solani* AG8 soon after the postinoculation period [31].

Comparing the data of all treatments and *PR-3*, the results were consistent for *GST* and *PAL*. This suggests that the material used for the experiments was appropriate, and that it was treated equally from cultivation to harvest, and from storage to sampling. Consistent basic gene expression levels were revealed for *PR-1*, *PR-3*, and *PI2* within the time-frame in all experiments. *PAL* and *GST* also showed consistently higher expression levels in experiments. *PAL* is a key enzyme in the phenylpropanoid pathway which is responsible for the biosynthesis of phytoalexins [32]. Therefore, it is proposed that SA synthesis via *PAL* occurs in cells at the infection site, thereby limiting fungal growth (Figure 5).

### 3.4. Antifungal Activities of Nanoparticles

The specific mechanism for the antifungal activity of NPs is still unknown. Some reports have investigated the electrostatic attraction between the negatively-charged cell membranes of microorganisms and positively-charged NPs, which is very important for the antibacterial regime of these particles [33]. However, in case of silver, it is suggested that NPs with large surface areas can quickly produce Ag^+^ by binding to the sulfhydral (−SH) practical groups of proteins and, consequently, denaturing proteins [34]. Silver NPs can also cause denaturation and destruction of proton pumps by binding to fungal surface proteins, increasing membrane permeability, or lipid bilayer proteins, which ultimately leads to the disruption of the cell membrane [35].

Concerning the effects of chitosan-based nanomaterials (NMs) on various fungal pathogens, interactions between chitosan NP molecules and the polyanionic structure of microbial cell membranes likely destabilize ethe cell membrane, resulting in the leakage of intracellular content and, ultimately, the death of the pathogen. Impaired protein synthesis and membrane destabilization are likely the primary and secondary modes of chitosan antimicrobial activity [36]. Although we obtained results which were in accordance with our expectations, the mechanism of the current nanocomposite may be far more complicated than assumed; future studies must endeavor to clarify this mechanism (Figure 7).

## 4. Materials and Methods

### 4.1. Isolation and Molecular Identification of R. solani and Tomato Variety Used in the Study

*Rhizoctonia solani* (GenBank Accession No.) was isolated from infected tomato plants [37]. Fungal spore cultures of the pathogen were purified and kept on potato dextrose agar (PDA) media and stored at 4 °C until further bioassay. Tomato (*Solanumly copersicum* L.) seeds of (Super strain B) variety were obtained from the Ministry of Agriculture, Egypt. Thirty sterilized conical flasks (250 mL) containing PD broth were seeded with ten selected fungal isolates (three repeats for each isolate) and incubated at 28 ± 2 °C. After 5 to 7 days of fungal inoculation on PDA media, approximately 100 mg of mycelial biomass were harvested [8]. The genomic DNA of each isolate was extracted using Biospin Fungus Genomic DNA Extraction Kit (Bioer, Hangzhou, China), following the manufacturer’s protocol. Purified DNAs were transferred into new tubes and stored at −20 °C until processing. The ITS region in the rDNA repeat of the 28S gene was amplified using primer (Appendix A) [38]. PCR amplification was carried out in a thermocycler ABI Gene Amp 9700 (Applied Biosystems, Waltham, MA, USA) accordingly. The obtained PCR product of ITS1 and ITS4 regions were sequenced using ABI PRISMTM 3100 DNA sequencer (Applied Biosystems) and Big Dye terminator sequencing kit (Version 3.1, Applied Biosystems, USA).

### 4.2. Preparation and Characterization of Ag/CHI Nanocomposites

Chitosan was dissolved at 0.5% (*w*/*v*) with 1% (*v*/*v*) acetic acid (HOAc), raised to pH 4.6–4.8, and filtered by a pump as previously described [39]. The fabricated chitosan NP was collected by centrifugation at 9000× *g* for 30 min. the NPs were rinsed with deionized water and then freeze-dried for further analysis. For Silver NPs, about 0.84 g silver nitrate (AgNO_3_) was dissolved in 50 mLof deionized water and diluted further. Root extract (5 mL) was added to the solution after diluting. The solution was autoclaved at 121 °C and 0.2 MPa for 15 min [40]. Ag NPs were collected by centrifugation and washed with deionized water.

To obtain Ag/CHI NC, a solution of Ag NPs and CHI NPs was mixed by sonication for about 1 h. The mixture was purified by centrifugation at 15 °C and 3600 rpm for 30 min. Supernatants were discarded and the mixture was extensively rinsed with deionized water to remove any sodium hydroxide and then freeze-dried for further analysis [39]. After drying, characterization of AgNPs, CHI NPs, and AgNPs/CHI NPs composites were made by Fourier Transform Infrared (FTIR) Spectrophotometer (SHIMADZU, Columbia, MD, USA) and 2100 plus Transmission electron microscopy (JEOL, Tokyo, Japan) system.

### 4.3. In vitro Antifungal Activity of Ag/CHI NC

The antifungal activity of Ag/CHI nanocompositein vitro for inhibiting *R. solani* radial mycelial growth was used with agar plate technique [41] with slight modifications. Seven different concentrations of Ag/CHI nanocomposites were added in deionized (DI) water and ultra-sonication were performed for better dispersion by sonicating for 1 h to make stable aqueous suspensions, then liquefied in sterilized PD broth media to obtain final concentrations 25, 50, 75, 100, 125, 150, 175 ppm NC: were used by the addition of oil to the melted media (Appendix A and Appendix A). For positive control, Nystatin (5 μL/well) was used as standard positive fungicide PDA media. Sterile distilled water was used in the bioassays instead of essential oil as a negative control set, then inoculated at the center with a mycelial disc (0.6 cm diameter) taken from the margins of 4–6 days old *R. solani* culture. Three replicate plates were used for each treatment, then the Petri-dishes were incubated at 25 °C and the fungal colony diameter was measured daily for 7 days.

### 4.4. Preparation of R. solani Fungal Suspension and Soil Infestation

Sterilized and nonsterilized soils were infested according to a method similar to [20]. For the preparation of *R. solani* isolate suspension five discs (5 mm diameter) of mycelia agar plugs of 7 days old were taken from the PDA plate margins: sand (2:1 *v*/*v*) and 10 mL sterile water in 2 L flask, then incubated at 25 ± 1 °C for two weeks before mixing with the soil of *R. solani* inoculated experiments by a 2% ratio [42].

### 4.5. Greenhouse Experiments

Seeds of Tomato (*S. lycopersicum*) were surface sterilized in sodium hypochlorite for 30 min, washed five times in sterile water, and germinated in peat moss for 15 d (irrigated regularly with H_2_O) and subsequently moved to pots experiment one plant per plastic pot of 18 cm diameter filled with sterile sandy-clay soil at 0.8 kg per pot and were arranged in a randomized complete block design with five replications and regularly irrigated with ¼ strength Hoagland solution as necessary and kept under natural daylight and humidity 65% until the end of each experiment. In the first pots group, the plants were under control treatment and regularly irrigated (C). In the second experiment, plants were under soil inoculated with *R. solani* fungal suspension one week before the transplanting process and regularly irrigated (P) for the next two weeks. In the third experiment, plants under control and regularly irrigated (c) were treated after transplantation with foliar of nano-fertilizer with Ag/CHI NC solution (50 mL) twice a day for three days (NC). In the fourth experiment, pots inoculated with *R. solani* were treated after transplantation with foliar of NFs with Ag/CHI NC solution (50 mL) twice a day for three days (P + NC). All plants continued growth with regular irrigation for two weeks after transplantation every 3 d for 2 weeks in a greenhouse at 22/16 °C, 65–70% humidity, and treatment and germination schedule presented in (Appendix A). All pots were evaluated for the incidence of *R. solani* root rot and stem rot.

### 4.6. Disease Assessments

Disease severity (DS) and incidence (DI) of *R. solani* root rot were assessed. Disease severity was evaluated using the 0–5 scale [43].
Disease severity (%) =Σab/AK × 100(1)
where, a = number of diseased plants with the same infection degree, b = infection degree, A = total number of the evaluated plants, and K = the greatest infection degree.

Disease incidence was calculated for each treatment according to the following Equation (2):Disease incidence (%) = a/A × 100(2)
where, a = number of diseased plants, and A = total number of evaluated plants.

### 4.7. Measurement of Plant Growth Parameters and Chlorophyll Contents

After sowing, the morphological traits of treated and untreated tomato plants were measured after 15 and 30 days of tomato seedlings. Three plants of each experiment were harvested for measuring plant height, leaf area, shoot and root fresh weight, shoot and root dry weight were measured after oven drying at 40 °C for 48 h.

Total chlorophyll content and anthocyanin level were measured on tomato plant leave after 30 days. Chlorophyll content was analyzed according to the method of [44], the pigments were extracted and grounded from 0.5 g of third fully expanded plant leaf between 8:00 and 10:00 am, suspended in 10 mL of 80% (*v*/*v*) acetone in the dark using a pestle and mortar. Extracts were filtrated and content of total Chll was determined by spectrophotometry at 645 and 663 nm. The anthocyanin level was measured using 0.5 g of leaves sample and soaked in 3 mL of acidified methanol (1% *v*/*v* HCl) for 12 h in darkness at 4 °C with occasional shaking. The mixture was centrifuged for 10 min at 14,000 rpm at 4 °C. The absorption of the extracts was estimated spectrophotometrically at 530 and 657 nm. Electrolytes leakage followed the methodology of [45].

### 4.8. Determination of TPC, TFC, and MDA Contents

The total phenolic content (TPC) of 30 days seedlings were prepared by dissolving 4.3 mg of air-dried plant powder in 10 mL methanol, according to [46]. The mixture was sonicated for 5 min to obtain a homogenized solution. To 300 μL of this solution taken in a test tube, 1 mL methanol, 3.16 mL distilled water, and 200 μL Folin-Ciocalteu reagents was added. Then, after 8 min incubation at room temperature, 600 μL sodium carbonate solutions (10%) were added and the test tube was covered with aluminum foil and incubated in a hot water bath at 40 °C for 30 min. The absorbance of the sample was determined using a UV visible spectrophotometer at 765 nm using UV-VIS spectrometer (Jenway, Tokyo, Japan).

Total flavonoid content (TFC) of tomato was studied using the aluminum chloride colorimetry method described by [47] with minor modifications. A standard calibration curve was constructed using quercetin in different concentrations (0.05−1 mg/mL). Tomato extract (2 mL) was mixed with 500 μL of 10% AlCl_3_ solution and 500 μL of 0.1 mM NaNO_3_ solution. After incubation at room temperature for 30 min, the absorbance of the reaction mixture was measured at the wavelength of 430 nm using UV-VIS spectrometer (Jenway, Japan).

Content of soluble protein was estimated in tomato plant following [48] using Folin phenol reagent and absorbance was recorded at 700 nm. Malondialdehyde (MDA) content in fresh tomato leaves was measured according to the method described by [49]. Briefly, 0.5 leaf samples were homogenized with 10 mL ethanol and followed centrifugation (10,000× *g*) for 10 min. The enzyme extract (1 mL) was added to 2 mL mixture of thiobarbituric acid (TBA, 0.65%) in trichloroacetic acid (TCA, 20%). The mixture was boiled for 30 min and then cooled rapidly. After centrifugation (10,000× *g*) for 5 min, the MDA contents were determined from the difference in nonspecific absorption at 600 and 532 nm.

### 4.9. Assay of Antioxidant Enzymes

Antioxidant enzymes were extracted by homogenizing 1 gm fresh tomato leaf tissue in chilled 50 mM phosphate buffer (pH 7.0) supplemented with 1% polyvinyl pyrolidine and 1 mM EDTA using prechilled pestle and mortar. After centrifuging the homogenate at 15,000× *g* for 20 min at 4 °C, the supernatant was used as an enzyme source.

The method described by [50] was used to determine superoxide dismutase activity (SOD, EC 1.15.1.1). For catalase assay (CAT, EC1.11.1.6) activity method of [51] was followed and the change in absorbance was monitored at 240 nm for 2 min. For calculation, an extinction coefficient of 39.4 mM^−1^cm^−1^ was used. Ascorbate peroxidase (APX, EC 1.11.1.11) activity was tested by monitoring absorption change at 290 nm for 3 min in 1 mL reaction mixture containing potassium phosphate buffer (pH 7.0), 0.5 mM ascorbic acid, hydrogen peroxide, and enzyme extract. The calculation of the extinction coefficient of 2, 8 mM^−1^ cm^−1^ was used [52].

### 4.10. Gene Expression

Total RNA was isolated from 0.5 g tomato plant root of all treatments by using the Plant RNA Kit (Sigma-Aldrich, Schnelldorf, Germany) according to the manufacturer’s protocol. The purified RNA was analyzed on 1% agarose gel and was reverse transcribed with reverse transcriptase (Promega, Germany) as previously described [53]. Quantitative Real-time PCR was carried out on 1 μL diluted cDNA by triplicate using the real-time analysis (Rotor-Gene 6000, Qiagen Corbett, Hilden, Germany). Primer sequences used in qRT-PCR were given (Appendix A). For Gene expression [54] method was used.

### 4.11. Statistical Analysis

Data Procession System (DPS) (Zhejiang University, China) was used for the analysis of variance (ANOVA). Correlation analysis and principal component analysis (PCA) were performed using the R version (3.4.2). Figures were constructed using software Origin (Origin Pro 9.0 for Windows).

## 5. Conclusions

In conclusion, this study provides new information about fungal infestation as it relates to potential risks and diseases caused by *Rhizoctonia solani* in tomato plants. The obtained results highlight the importance of future research to control the occurrence of *R. solani* in tomatoes. Although significant upregulation of defense-related genes was observed, increased antioxidants, TFC, and TPC values suggest a biological mechanism which controls elevated ROS levels due to following infection. Furthermore, our results indicated that Ag-Chit-NCs should be further investigated as effective fungicides for applications in agriculture and food safety. The use of the proposed NCs in the form of a disinfectant spray could successfully prevent food contamination caused by fungi.

## Figures and Tables

**Figure 1 plants-10-02283-f001:**
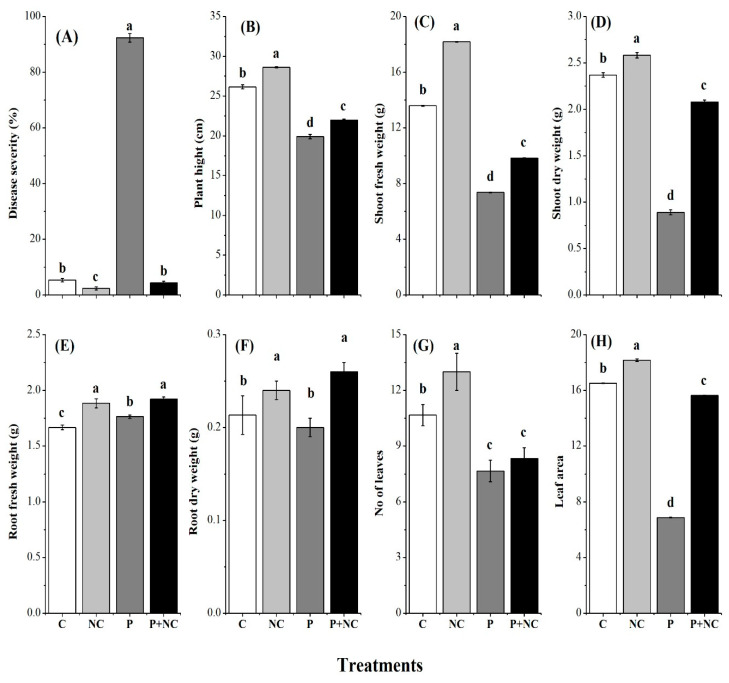
Different physical parameters of plant after treatments. Disease severity (DS) (**A**), Plant height (PH) (**B**), Shoot fresh weight (SFW) (**C**), Shoot dry weight (SDW) (**D**), Root fresh weight (**E**), Root dry weight (**F**), No of leaves (**G**), Leaf area (**H**). Different lower-case letters indicate significant difference (*p* ≤ 0.01) among the different treatments. Error bars indicate ± standard error of the mean of three replicates. Note: C for control, NC = plants treated with nano-fertilizer with Ag/CHI NC, P = plants treated with *R. solani*, (P + NC) = pots inoculated with *R. solani* and Ag/CHI NC solution (50 mL) twice a day for three days.

**Figure 2 plants-10-02283-f002:**
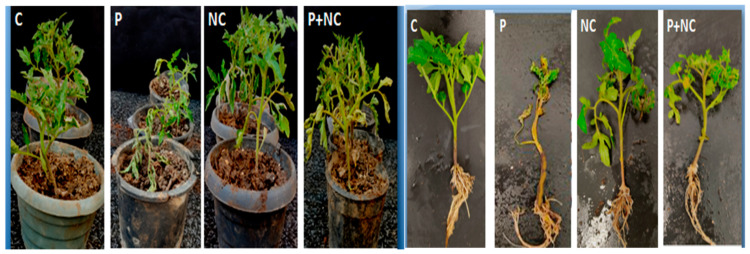
Physical appearance of plant after treatments. Left; Tomato (*Solanum lycopersicum*) plant after 14 days treatments in pot experiment, Right;plant morphology at the end of pot experiments after 30 days. Note: C for control, NC = plants treated with nano-fertilizer with Ag/CHI NC, P = plants treated with *R. solani*, (P + NC) = pots inoculated with *R. solani* and Ag/CHI NC solution (50 mL) twice a day for three days.

**Figure 3 plants-10-02283-f003:**
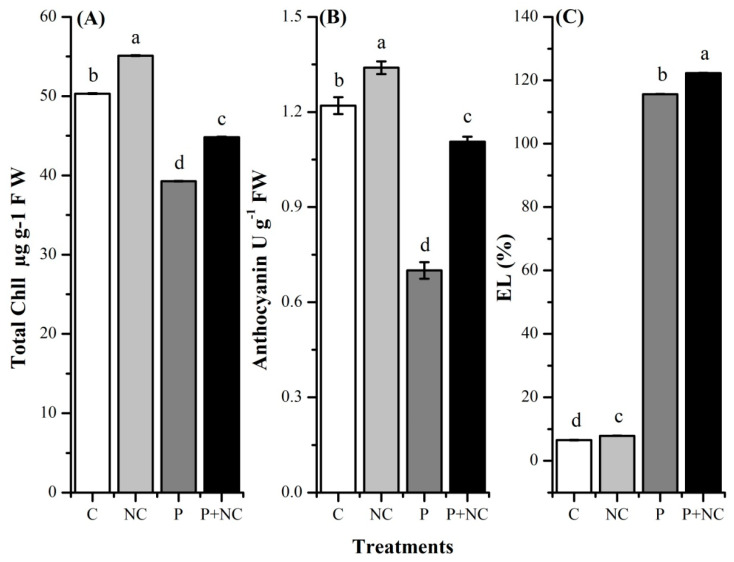
Different physiological parameters of plant after treatments. Total Chll (**A**), Anthocyanin (**B**), and electrolyte leakage (EL) (**C**). Different lower-case letters indicate significant difference (*p* ≤ 0.01) among the different treatments. Error bars indicate ± standard error of the mean of three replicates. Note: C for control, NC = plants treated with nano-fertilizer with Ag/CHI NC, P = plants treated with *R. solani*, (P + NC) = pots inoculated with *R. solani* and Ag/CHI NC solution (50 mL) twice a day for three days.

**Figure 4 plants-10-02283-f004:**
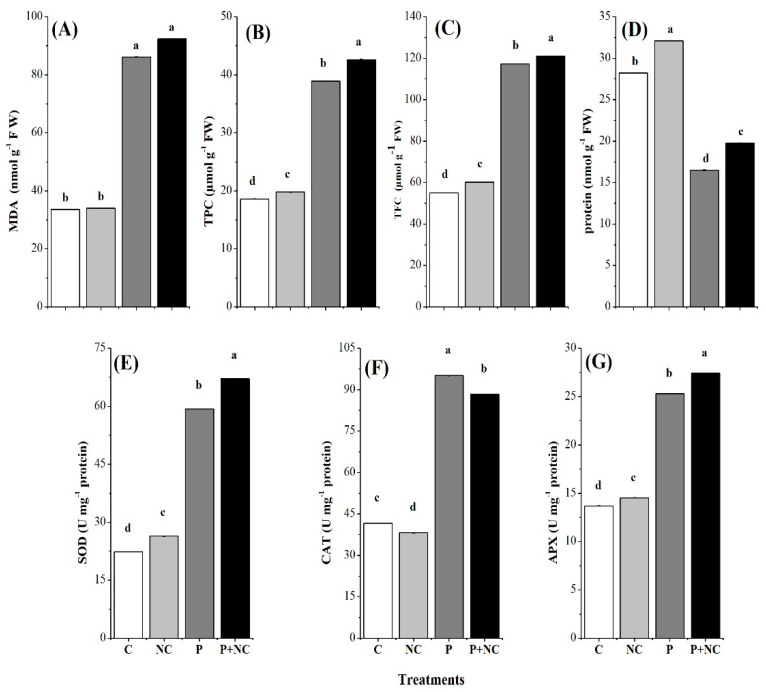
Different biochemical parameters of plant after treatments. Malondialdehyde MDA (**A**), Total phenolic contents (TPC) (**B**), Total flavonoid content (TFC) (**C**), Total Protein (**D**), Superoxide Dismutase (SOD) (**E**), Catalase (CAT) (**F**), and Ascorbate Peroxidase (APX) (**G**). Different lower-case letters indicate significant difference (*p* ≤ 0.01) among the different treatments. Error bars indicate ± standard error of the mean of three replicates. Note: C for control, NC = plants treated with nano-fertilizer with Ag/CHI NC, P = plants treated with *R. solani*, (P + NC) = pots inoculated with *R. solani* and Ag/CHI NC solution (50 mL) twice a day for three days.

**Figure 5 plants-10-02283-f005:**
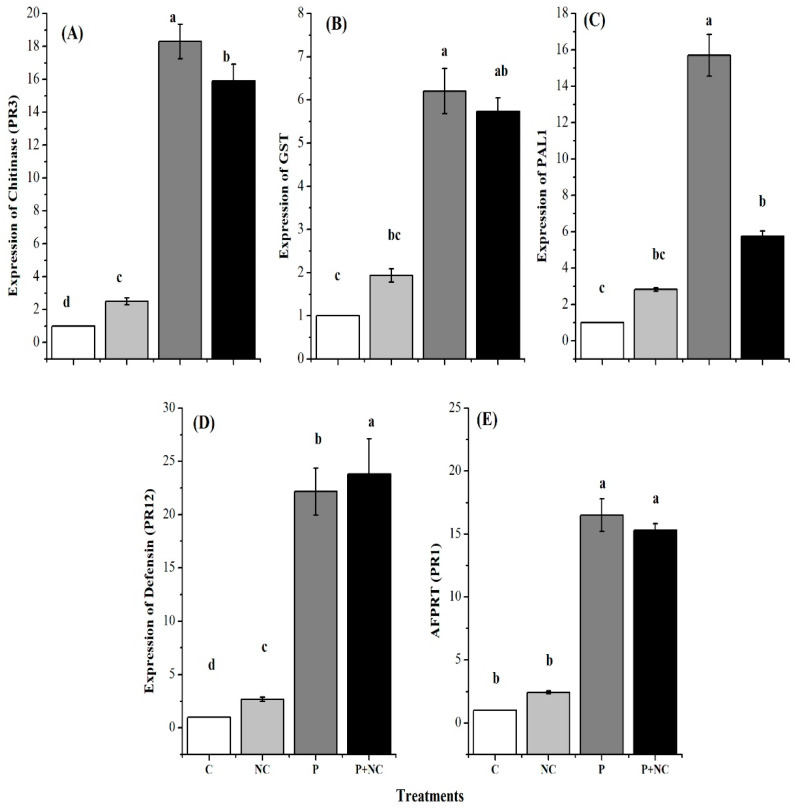
Different plant defense related genes expression after treatments. Expression of Chitinase (**A**), Glutathione-S-transferase (*GST*) (**B**), Phenyl ammonia lyase *PAL1* (**C**), *Defensin* (**D**), and *Pathogenesis-related protein* (*AFPRT*) (**E**). Different lower-case letters indicate significant difference (*p* ≤ 0.01) among the different treatments. Error bars indicate ± standard error of the mean of three replicates. Note: C for control, NC = plants treated with nano-fertilizer with Ag/CHI NC, P = plants treated with *R. solani*, (P + NC) = pots inoculated with *R. solani* and Ag/CHI NC solution (50 mL) twice a day for three days.

**Figure 6 plants-10-02283-f006:**
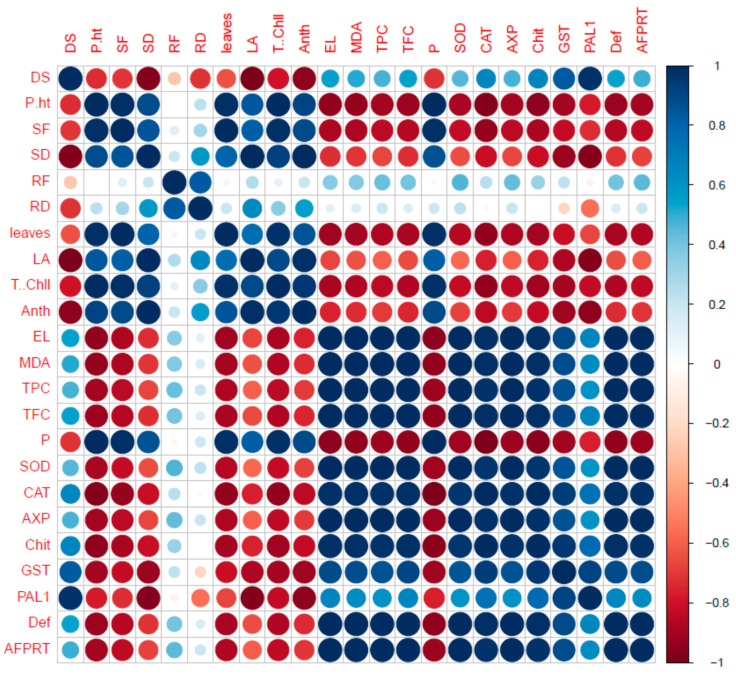
Correlation analysis among different parameters plant grown under different treatments.

**Figure 7 plants-10-02283-f007:**
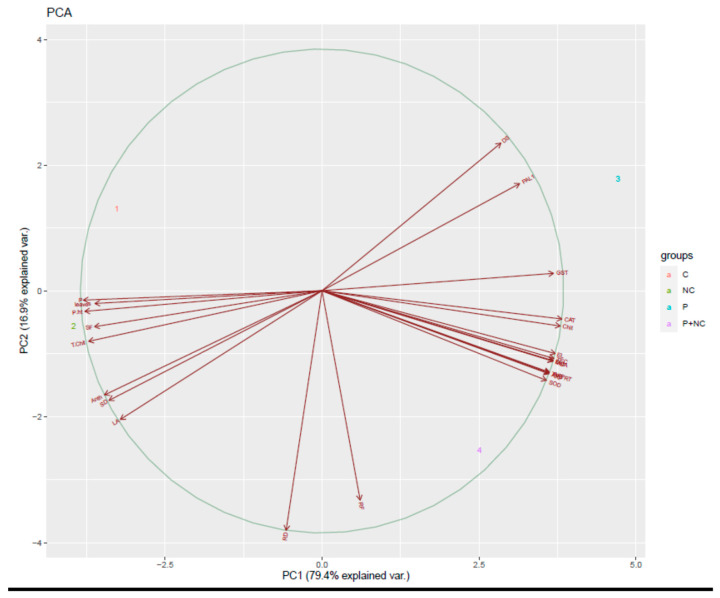
Principal component analysis (PCA) of plant grown under different treatments.

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
