# Peer review of "The Antifungal Activity of Ag/CHI NPs against Rhizoctonia solani Linked with Tomato Plant Health"

_plants, 2021, doi:10.3390/plants10112283_

Round 1

Reviewer 1 Report

Authors showed that inoculation of tomato plants with the pathogenic fungus (P variant) resulted in the disease development up to 90% compared to the control (C) and experimental (NC) variants. In the case of the P + NC variant, the level of disease development was the same as in the control (C). Thus, treatment of tomato plants with the nano-fertilizer  with Ag/CHI NC solution (50 ml) provided a protection effect against the disease caused by Rhizoctonia solani. Fungal infection of plants reduced their height as well as the fresh and dry weight of their stems that indicates the effect of the infection on the physiological processes occurring in the aboveground parts of tomato plants. This infection also significantly influenced on the average number of leaves per a plant and the average leaf area. At the same time, it did not significantly influence on the fresh and dry weight of roots.  In my opinion, the submitted manuscript meets the scope of the ‘Plants’ journal.

However, this manuscript needs to be significantly improved to be suitable for publication.

One of the main weaknesses of this study is its non-readability. The text, including the title, is confusing and contains many unclear and confusing sentences. Many phrases cannot be understood because of errors and wrong choice of words. The authors should revise the whole text. They must correct numerous grammar mistakes, misprints and merged words. I would recommend authors to use the services of a native English speaker or/and professional proof-reading and to rewrite the manuscript in order to provide the better understanding of what they want to say.

Some specific comments and suggestions

  1. The full explanation of abbreviations should be given where they first appear in both the abstract and the main text. Full names of some terms are excessively repeated several times (e.g., lines 133-134 and 146-147), while other names (e.g. MDA, EL) are not explained or are clarified in Mat & Meth., but not at the places of their first mentioning (e.g., PH, line 131).
  2. Figures should be supplemented with more expanded explanation below captions. The general rule is that results illustrated by figures must be completely understandable to the reader independently of the main text. For instance, the explanation of such abbreviations as C, P, NC, etc. should be given not only in the text, but also under Figure 1.
  3. Error bars (SE), which are mentioned in the captions of Figs. 4 & 5, are not indicated on the diagram for the corresponding treatments.
  4. Lines 69-70

Probably, the worst tomato pathogen just for some regions? Actually, among fungal diseases impacting tomato crops worldwide, the early blight (Alternaria alternata, A. solani) is considered as the most devastating disease.

Please replace this reference (Heflish et al., 2017) with one dedicated to the tomato damage caused by R. solani, since this reference does not include this information.

  1. Latin names of the pathogen and gene abbreviations must be written in italic. Check and correct across the whole text.

Some (but not all) examples of the expression errors and grammar mistakes.

line 42-43

The phrase is confusing and should be reformulated. Do you mean that expression of genes encoding the listed pathogenesis-related proteins of tomato was evaluated, and a significant difference in the expression levels in treated and control plants was revealed in relation to the housekeeping gene? The housekeeping gene should be indicated here or in Mat. & Meth.

lines 46 and 265

Why 'astonishing'? Author should add more specific elaboration on the novelty of this article. Antimicrobial activity of Ag is well-known. CHI is known as an active elicitor of plant defense responses, the use of which results in the reduction of various diseases including tomato ones.

lines 71-72

''..these pathogens...'' - a confusion is here. If you mean R. solani, "these'' is a grammar mistake. If you mean above-mentioned plant pathogens, it is incorrectly to use "these'' since fungicide application is not the main way to combat viruses, bacteria, and nematodes.

line 81

Please change ‘necrotic bacteria’ to ‘necrotrophic fungi and bacteria’.

line 87

This needs to be corrected. PR-1 and PR-1 expression are, respectively, the marker protein and a marker for SA-dependent signaling pathway or SAR, but not markers of salicylic acid itself. Line 399 – the same problem.

line 95

The corresponding references are needed to be added.

line 129

If (P) means «fungus applied treatment», than there is a clear tautology in the phrase "the P treatment" (fungus applied treatment treatment).

line 177-178

The phrase should be revised. … The second part of this study was the biochemical analyses of the fungal  treated plants… and its influence on MDA, total phenolic contents (TPC), total flavonoid contents (TFC), and other Proteins of the plants.

– its (?) The influence of the study?!

– A wrong use of ‘other’. Phenols and flavonoids are not proteins.

– Proteins (?) Do you mean total protein content?

line 192

Phenolic and flavonoid compounds but not ‘total content’ of these plant metabolites.

line 448

What does NM mean?

line 457 

Fig. 1A includes the "-20%" value on the Y axis. This corresponds to the disease development level equal to -20%, what does it mean??

lines 575 - 578  

(NC)  In the third experiment, "plants under control  and regularly irrigated" were treated after transplantation with foliar of nano-fertilizer  with Ag/CHI NC solution (50 ml ) two times one every three days.  - What did you mean by the last phrase (especially by the "one" word? they were treated two times every three days? Please, explain.

In the fourth  experiment, pots inoculated with R. solani were treated after transplantation with foliar  of NFs with Ag/CHI NC solution (50 ml) two times one every three days (P+NC).   - The same question.

Authors showed that inoculation of tomato plants with the pathogenic fungus (P variant) resulted in the disease development up to 90% compared to the control (C) and experimental (NC) variants. In the case of the P + NC variant, the level of disease development was the same as in the control (C). Thus, treatment of tomato plants with the nano-fertilizer  with Ag/CHI NC solution (50 ml) provided a protection effect against the disease caused by Rhizoctonia solani. Fungal infection of plants reduced their height as well as the fresh and dry weight of their stems that indicates the effect of the infection on the physiological processes occurring in the aboveground parts of tomato plants. This infection also significantly influenced on the average number of leaves per a plant and the average leaf area. At the same time, it did not significantly influence on the fresh and dry weight of roots.  In my opinion, the submitted manuscript meets the scope of the ‘Plants’ journal.

However, this manuscript needs to be significantly improved to be suitable for publication.

One of the main weaknesses of this study is its non-readability. The text, including the title, is confusing and contains many unclear and confusing sentences. Many phrases cannot be understood because of errors and wrong choice of words. The authors should revise the whole text. They must correct numerous grammar mistakes, misprints and merged words. I would recommend authors to use the services of a native English speaker or/and professional proof-reading and to rewrite the manuscript in order to provide the better understanding of what they want to say.

Some specific comments and suggestions

  1. The full explanation of abbreviations should be given where they first appear in both the abstract and the main text. Full names of some terms are excessively repeated several times (e.g., lines 133-134 and 146-147), while other names (e.g. MDA, EL) are not explained or are clarified in Mat & Meth., but not at the places of their first mentioning (e.g., PH, line 131).
  2. Figures should be supplemented with more expanded explanation below captions. The general rule is that results illustrated by figures must be completely understandable to the reader independently of the main text. For instance, the explanation of such abbreviations as C, P, NC, etc. should be given not only in the text, but also under Figure 1.
  3. Error bars (SE), which are mentioned in the captions of Figs. 4 & 5, are not indicated on the diagram for the corresponding treatments.
  4. Lines 69-70

Probably, the worst tomato pathogen just for some regions? Actually, among fungal diseases impacting tomato crops worldwide, the early blight (Alternaria alternata, A. solani) is considered as the most devastating disease.

Please replace this reference (Heflish et al., 2017) with one dedicated to the tomato damage caused by R. solani, since this reference does not include this information.

  1. Latin names of the pathogen and gene abbreviations must be written in italic. Check and correct across the whole text.

Some (but not all) examples of the expression errors and grammar mistakes.

line 42-43

The phrase is confusing and should be reformulated. Do you mean that expression of genes encoding the listed pathogenesis-related proteins of tomato was evaluated, and a significant difference in the expression levels in treated and control plants was revealed in relation to the housekeeping gene? The housekeeping gene should be indicated here or in Mat. & Meth.

lines 46 and 265

Why 'astonishing'? Author should add more specific elaboration on the novelty of this article. Antimicrobial activity of Ag is well-known. CHI is known as an active elicitor of plant defense responses, the use of which results in the reduction of various diseases including tomato ones.

lines 71-72

''..these pathogens...'' - a confusion is here. If you mean R. solani, "these'' is a grammar mistake. If you mean above-mentioned plant pathogens, it is incorrectly to use "these'' since fungicide application is not the main way to combat viruses, bacteria, and nematodes.

line 81

Please change ‘necrotic bacteria’ to ‘necrotrophic fungi and bacteria’.

line 87

This needs to be corrected. PR-1 and PR-1 expression are, respectively, the marker protein and a marker for SA-dependent signaling pathway or SAR, but not markers of salicylic acid itself. Line 399 – the same problem.

line 95

The corresponding references are needed to be added.

line 129

If (P) means «fungus applied treatment», than there is a clear tautology in the phrase "the P treatment" (fungus applied treatment treatment).

line 177-178

The phrase should be revised. … The second part of this study was the biochemical analyses of the fungal  treated plants… and its influence on MDA, total phenolic contents (TPC), total flavonoid contents (TFC), and other Proteins of the plants.

– its (?) The influence of the study?!

– A wrong use of ‘other’. Phenols and flavonoids are not proteins.

– Proteins (?) Do you mean total protein content?

line 192

Phenolic and flavonoid compounds but not ‘total content’ of these plant metabolites.

line 448

What does NM mean?

line 457 

Fig. 1A includes the "-20%" value on the Y axis. This corresponds to the disease development level equal to -20%, what does it mean??

lines 575 - 578  

(NC)  In the third experiment, "plants under control  and regularly irrigated" were treated after transplantation with foliar of nano-fertilizer  with Ag/CHI NC solution (50 ml ) two times one every three days.  - What did you mean by the last phrase (especially by the "one" word? they were treated two times every three days? Please, explain.

In the fourth  experiment, pots inoculated with R. solani were treated after transplantation with foliar  of NFs with Ag/CHI NC solution (50 ml) two times one every three days (P+NC).   - The same question.

Author Response

Author: Thank you for your constructive comments. We have carefully considered the advice and revised our manuscript accordingly. We hope that this revised manuscript has been much improved.

Reviewer 2 Report

General comments

This study assesses the impact of a fungal pathogen, Rhizoctonia solani, on tomato plants over a time course of infection. Measurement of changes to plant content (e.g., chlorophyll and anthocyanin) are reported. The connection with nanoparticles is not clear in the abstract and many terms in the abstract are introduced making this section difficult to follow. The flow of the introduction needs to be improved, as detail about disease is provided but a connection to the application of nanoparticles is not clear and a purpose of the study, as well as direction and findings should be presented. I have serious concerns about the presentation of the study and flow of information is very difficult to follow. I have provided some early comments below but the manuscript needs to be revised for comprehension to warrant a more in-depth review.

Specific comments

Abstracts: watch italics of scientific name

  • Define TPC, TFC, MDA, CAT, SOD, APX, ROS, TEM, FTIR, NP, etc. or provide context, as appropriate.

Introduction: at times, disjointed. What is the connection between antifungal activity, plant health, defense, and soil infestation? Lots of information provide as background, but a paragraph at the end of the Introduction bringing everything together and why it’s relevant for the study would be helpful to the reader.

Results: duplication of information presented in introduction not needed (line 123).

Difficult to understand the message (e.g., line 127). What is fungus applied treatment and how do the levels relate to disease severity?

Author Response

“Please see attachment” 

Round 2

Reviewer 1 Report

Authors showed that inoculation of tomato plants with the pathogenic fungus (P variant) resulted in the disease development up to 90% compared to the control (C) and experimental (NC) variants. In the case of the P + NC variant, the level of disease development was the same as in the control (C). Thus, treatment of tomato plants with the nano-fertilizer  with Ag/CHI NC solution (50 ml) provided a protection effect against the disease caused by Rhizoctonia solani. Fungal infection of plants reduced their height as well as the fresh and dry weight of their stems that indicates the effect of the infection on the physiological processes occurring in the aboveground parts of tomato plants. This infection also significantly influenced on the average number of leaves per a plant and the average leaf area. At the same time, it did not significantly influence on the fresh and dry weight of roots.  In my opinion, the submitted manuscript meets the scope of the ‘Plants’ journal.

Author Response

"I think data presentation is good and and flow of the manuscript is better.
I believe that after minor editing (spelling at some fragments, spaces
between fragments) it can be published. "

Author: Thanks. We have revised the manuscript and corrected spellings as well as the problems of the space.